# Caregivers’ Willingness to Vaccinate Their Children against Childhood Diseases and Human Papillomavirus: A Cross-Sectional Study on Vaccine Hesitancy in Malawi

**DOI:** 10.3390/vaccines9111231

**Published:** 2021-10-22

**Authors:** Gbadebo Collins Adeyanju, Philipp Sprengholz, Cornelia Betsch, Tene-Alima Essoh

**Affiliations:** 1Psychology and Infectious Disease Lab (PIDI), Media and Communication Science, University of Erfurt, Nordhäuser Straße 63, 99089 Erfurt, Germany; philipp.sprengholz@uni-erfurt.de (P.S.); cornelia.betsch@uni-erfurt.de (C.B.); 2Centre for Empirical Research in Economics and Behavioral Science (CEREB), University of Erfurt, Nordhäuser Straße 63, 99089 Erfurt, Germany; 3Agence de Médecine Préventive, Regional Directorate for Africa, Abidjan 08 BP 660, Côte d’Ivoire; tae@aamp.org

**Keywords:** immunization, vaccine hesitancy, vaccine demand, vaccination, Malawi, drivers, caregivers, childhood diseases, human papillomavirus (HPV), adolescent

## Abstract

Background: Vaccines are among the most effective and cost-efficient public health interventions for promoting child health. However, uptake is considerably affected by vaccine hesitancy. An example is Malawi, with a decline in second vaccine doses and the highest cervical cancer incidence and mortality rate in Sub-Saharan Africa. Understanding vaccine hesitancy is especially important when new vaccines are introduced. This study explores factors contributing to vaccine hesitancy for routine childhood immunization and the human papillomavirus vaccine in Malawi. Methods: The study used a cross-sectional survey design targeting caregivers of children under five years old and adolescent girls. The sample population was derived using three inclusion criteria: one district with low vaccine uptake (Dowa), one district with high vaccine uptake (Salima), and one district where human papillomavirus vaccine was piloted earlier (Zomba). A convenience sample of one primary and one secondary health facility was selected within each district, and participants were systematically included (*n* = 600). The measures were based on 5C scale for measuring vaccine hesitancy. Multiple regression analyses were performed to explore vaccination intention predictors. Results: Confidence in vaccine safety was the strongest predictor of routine childhood immunization, followed by constraints due to everyday stress. Caregivers had lower confidence in vaccine safety and efficacy when they believed rumors and misinformation and were unemployed. Confidence was higher for those who had more trust in healthcare workers. Age, gender, religion, education, employment, belief in rumors, and trust in healthcare workers were considered predictors of vaccination intention. A husband’s positive attitude (approval) increased childhood vaccination intention. For human papillomavirus, vaccination intentions were higher for those with lower education, more trust in healthcare workers, lower complacency, and a lower tendency toward calculating the benefits and risks of vaccination. Knowledge of human papillomavirus did not increase vaccination intention, but the need to attain a husband’s approval did. Being a young adult and unemployed increased belief in rumors, while trust in healthcare workers reduced the belief. Conclusions: This study provides good insights into the drivers of vaccine hesitancy across different contexts in Malawi. However, further studies are necessary to understand low risk perception among elderly people and the declining trend in second vaccine doses.

## 1. Introduction

Vaccination is regarded as one of the most effective and cost-effective public health innovations for promoting child health because of its direct health benefits and positive externalities [1]. It can prevent morbidity and mortality from vaccine-preventable diseases and contribute to national disease elimination and eradication efforts [2]. The World Health Organization (WHO) and the United Nations Children’s Emergency Fund (UNICEF) estimate that vaccination prevents about two to three million deaths in children annually [3]. In addition, remarkable progress has been made toward global polio eradication due to global vaccination programs and efforts [3,4]. To ensure sustainable and equitable access to vaccines, countries coordinate immunization-related activities within global and national immunization programs. The global indicator to measure vaccine uptake is the coverage rate with the third dose of the diphtheria-tetanus-pertussis-containing vaccine (DTP3), which is expected to be 90% at the national level and 80% at the district level [5,6]. However, in 2018, the global average for DTP3 coverage was about 86% [7].

Between 2000 and 2019, over 822 million children were immunized worldwide, 1.1 billion vaccinations were supported via multiple campaigns, an estimated 14 million deaths were averted, and USD 150 billion economic benefits were generated due to immunization [8]. An additional 300 million children were to be immunized against potentially fatal diseases by the end of 2020, saving between five and six million lives, preventing 250 million disability-adjusted life years, and reducing under-five mortality by 10% [9,10].

Despite this considerable progress, one in five children globally remains unvaccinated or partially vaccinated, which contributes to about 1.5 million deaths from vaccine-preventable diseases annually [11,12,13]. In the WHO African region, DTP3 coverage has stagnated at 76% [14], and the region contribute the highest proportion of under-vaccinated and consequent child deaths from vaccine-preventable diseases globally [15,16,17]. Of the 10 countries that account for 11.7 (60%) of the 19.7 million non- or under-vaccinated children globally, 40% are in Sub-Saharan Africa (SSA), including Nigeria, Ethiopia, Democratic Republic of Congo, and Angola [18]. Average coverage rates in the SSA region remain sub-optimal, stagnating over the past five years at 72% [19,20]. Despite the promises that vaccines exhume, recent data have shown a different and negative behavior to vaccination uptake, which can be regarded as vaccine hesitancy [21,22]. 

In Malawi, the percentage of fully immunized children aged 12–23 months has been declining since 1992, when coverage peaked at 82% [23]. This declined to 64% in 2004, then rose to 81% and 99% in 2010 and 2012, respectively [23]. In 2018, the national average vaccination coverage rates for three doses of DTP3 in Malawi declined again to 92% [24].

A study on vaccination coverage and timeliness with valid doses in Malawi showed that, while the availability of vaccination cards (evidence of inoculation) in the Dowa and Ntchisi districts was as high as 94% and vaccination coverage by card and mothers’ history was also as high as 93% for all antigens, the percentages of valid doses completed by children was 60% in Dowa District and as low as 49% in Ntchisi District [25]. The assessment showed that many children in the two districts had an incomplete number of doses.

### 1.1. Cervical Cancer Burden and HPV Vaccination

Cervical cancer will kill more than 443,000 women per year worldwide by 2030, and nearly 90% of the deaths will be in SSA [26,27]. The burden of this disease is most severe in low- and middle-income countries. The top 20 countries with the highest burden of cervical cancer cases globally in 2018 were all in SSA, except for Bolivia [28,29]. Within SSA, East Africa has the highest incidence rate, while Malawi is among not just the global leaders but also top in the former [24,29]. Malawi has the second highest burden of cervical cancer globally and the highest in the SSA region [27]. It is the most common cancer in women in the country, accounting for 45.4% of all female cancer incidence [29,30]. Of all diagnosed cancers among women in Malawi, 80% will die prematurely [31,32]. Overall, about 5 million Malawian women aged 15–44 are most at risk of developing cervical cancer [33,34].

Cervical cancer is caused by the human papillomavirus (HPV), a double-stranded DNA virus [35]. It is one of the most prevalent sexually transmitted diseases, with over half of sexually active individuals contracting HPV during their life [36]. Despite the existence of over 200 HPV types, 70% HPV-related pathologies are due to infection with HPV types 16, 18, 6, and 11 [37,38]. 

The HPV vaccine, targeting girls between 9 and 13, was introduced into the routine immunization (RI) program in Malawi to reduce the high rate of cervical cancer deaths among women [39]. Through vaccination, cervical cancer can be prevented by vaccinating adolescent girls before they become sexually active, helping to reduce the spread of the virus and, consequently, lower cervical cancer mortality. Prior to its introduction into the RI program in Malawi in 2019, the pilot demonstration (2013–2016) successfully vaccinated 26,766 in-school girls and aimed to vaccinate 240,000 girls of 9 years old, before expanding in future to include 1.5 million adolescent girls aged 9–14 [40,41]. The pilot demonstration revealed a decline between the first and the second dose in the Rumphi (98% to 88%) and Zomba districts (89% to 76%) [42,43]. Despite the possibility of safe protection against cervical cancer and vaccine availability, uptake remained below 90% due to several factors, including vaccine hesitancy. 

### 1.2. Vaccine Hesitancy

Although many factors may be responsible for low childhood and HPV vaccine coverage, vaccine hesitancy is recognized as an important contributor [22,44], hence the underlining motive of this study. The Strategic Advisory Group of Experts (SAGE) on Immunization defines vaccine hesitancy as a “delay in acceptance or refusal of vaccines despite availability of vaccination services” [45,46]. This definition suggests that vaccine hesitancy is a demand-side problem that influences vaccination uptake because of several complex factors, including perception about vaccines, fear of adverse events, religious values, and a general lack of trust in healthcare professionals or the healthcare system [47].

Vaccine hesitancy ranges from delay in acceptance of vaccines to complete refusal. It is driven by factors such as confidence (level of trust in vaccine or provider), complacency (not perceiving a need for vaccine or not valuing the vaccine), and convenience (access) [47]. It is also context-specific, as it depends on time, place, the specific vaccine, and the societal context. Therefore, this study will examine context- and vaccine-specific determinants of vaccine hesitancy to inform context-specific strategies, especially for new and underutilized vaccines, such as the HPV vaccine. 

Even in SSA, where vaccination has been the hallmark of public health intervention for development in the last 40 years, vaccine hesitancy is causing vaccination uptake to slow down, stagnate, or even decrease [45,47,48,49,50,51]. In Malawi, the WHO/UNICEF Joint Reporting Form for 2018 named religious factors, perception, and lack of awareness as reasons for vaccine hesitancy [52]. However, the reporting was not grounded in evidence; instead, it relied on the opinions of field health officers [53]. The dearth of empirical evidence for understanding vaccine hesitancy has hindered robust and responsive intervention and, therefore, justified this study. 

A few studies have shown that caregivers who are hesitant about vaccination are more likely to attend vaccination appointments late [54]. Additionally, a study shows an association between hesitancy and missed opportunities for vaccination in Malawi [55]. In Malawi, 66% of children eligible for vaccination did not receive at least one vaccination despite availability; in addition, 92% of people attending health facilities for non-vaccination visits and who were eligible for vaccination had at least one missed opportunity, and 57% have missed more [56]. Caregivers of adolescent girls who refused the HPV vaccination for their daughters or who did not complete doses during the HPV demonstration project in the Rumphi and Zomba districts of Malawi named reasons such as inconvenient location and time, belief that the vaccine portends danger to the girls, and the vaccination site being unclean and not safe [40]. Education about cervical cancer, inadequate information about vaccination opportunities, fears of side effects, and a general distrust toward new vaccines were some of the identified factors driving vaccine hesitancy in the early introduction of the HPV vaccine to the country [57,58]. Generally, because of the target population (adolescent girls) and amplified by rumors, mistrust of HPV vaccination seems widespread in low-income settings [59,60,61]. 

Currently, vaccine hesitancy is galvanizing unprecedented scholarly focus globally in view of the COVID-19 pandemic; however, at the same time, there is a vacuum of knowledge, especially in the Africa region. This limits the extent of evidence-based intervention in the region. No scientific model has yet been explored in Malawi that measures vaccine hesitancy and compares the relative impact of these influencing factors. Hence, this study’s goal was to systematically explore factors that influence vaccine hesitancy in Malawi among caregivers of children and adolescent girls who are eligible for RI and HPV vaccination, respectively. Additionally, to assess the depth of vaccination knowledge among caregivers in Malawi. Multi-dimensional tools and ways to measure vaccine hesitancy exist, but little is known about their validity in non-Western settings, such as SSA [51]. Other factors may be relevant in SSA, but no tool currently exists to assess and extend existing measures. Therefore, this study assessed use of an expanded 5C psychological antecedents model to understand vaccine hesitancy drivers in Malawi. 

## 2. Methods

### 2.1. Design and Setting

The study used a cross-sectional study design and was initially planned to be conducted in Rumphi, Dowa, Zomba, and Nsanje districts. However, due to the SARS-CoV-2 pandemic situation in Malawi in April 2020, data were collected in Salima, Dowa, and Zomba districts, which were selected based on the following criteria: one district with high vaccine coverage (Salima at 91%), one district with low vaccine coverage (Dowa at 70%), and one district where the HPV vaccination had been piloted (Zomba) [43].

### 2.2. Study Population and Sample Size

The study population included caregivers of children under five years of age and caregivers of adolescent girls whose children were eligible for HPV vaccination at local health centers. A total sample of 800 was planned (*n* = 200 per district), but a reduced sample of 600 participants (*n* = 200 per district) was enrolled due to the pandemic situation. The fourth district (Nsanje) was canceled due to travel restrictions. 

### 2.3. Sampling and Data Collection

Within each stratum (district), a convenience sample of one primary and one secondary health facility was selected. For each sample, a systematic sampling technique was used to select study participants [62]: every third caregiver who visited the facility was included until sample saturation. Every caregiver who met the eligibility criteria was included until a sample of *n* = 200 was reached in each district. The data collection tool/questionnaire is available at the Open Science Framework https://osf.io/pzaer/ accessed on 20 October 2021. 

### 2.4. 5C+ Model for Measuring Vaccine Hesitancy

There are five psychological antecedents of vaccination behavior represented in the 5C model that measures vaccines hesitancy: confidence, complacency, constraints, calculation, and collective responsibility [49]. *Confidence* is trust in the effectiveness and safety of vaccines. *Complacency* exists where the perceived risks of vaccine-preventable diseases are low, and vaccination is not deemed a necessary preventive action. *Constraints* are an issue when physical availability, affordability, and willingness-to-pay, geographical accessibility, ability to understand (language and health literacy), and appeal of immunization service affect uptake. *Calculation* refers to individuals’ engagement in extensive information searching and should therefore be related to perceived vaccination and disease risks. *Collective responsibility* is the willingness to protect others by one’s own vaccination by means of herd immunity [49]. 

Religion, rumors, and masculinity were added to the set of items, referring to the augmented scale as “5C+.” *Religion* has been found to be an important factor affecting people’s attitudes toward vaccine demand [63,64,65]. Religious reasons for declining immunization reflect the role of beliefs among faith communities [49,63,66]. *Masculinity* is used here to connote a husband or father’s role in the household’s decision to vaccinate a child. A husband’s approval (attitude) for the child to be vaccinated plays an important role in vaccination acceptance and refusal. Caregivers who solely depend on their husband’s approval are prone to vaccinate less if the husband does not approve [67]. Finally, *rumors or misinformation* affects perceptions and everyday life, amplified in the age of social media, and this has been found to have some impact on vaccination demand in previous studies [68,69]. In addition, since *knowledge* is associated with HPV vaccination behavior, this variable was considered important in the assessment of vaccine hesitancy drivers in Malawi. 

### 2.5. Measures 

All participants completed a paper and pencil questionnaire. The study collected demographic information about the participants, including age, gender (female, male, other), religious affiliation (Christian, Muslim, other), and educational attainment (primary, secondary, tertiary, other). The questionnaire contained the following constructs and items: 

*Sources of vaccination information and trust levels:* We assessed the caregivers’ sources of vaccination information (healthcare workers, family members, religious groups or places of worship, social media, and community members). Trust in each of the above sources was measured with a rating scale from 1 = “Not at all” to 5 = “I totally trust.” 

*Knowledge of HPV/cervical cancer*: Knowledge of cervical cancer and HPV was tested with “Have you ever heard about cervical cancer and the HPV vaccine?”, “Have you heard of the virus that causes cervical cancer?”, and “Do you know that cervical cancer can be prevented?” These were answered with “yes” or “no.” 

*5C psychological antecedents of vaccination:* The 5C measured the confidence, complacency, constraints, calculation, and collective responsibility variables. Religion, rumor, and masculinity as stand-alone additional variables were similarly measured. All items were measured on a scale ranging from 1 = “strongly disagree” to 5 = “strongly agree”. The 5C+ were measured as follows: 

*Confidence*: “I am completely confident that vaccines are safe,” “Vaccinations are effective,” and “Regarding childhood vaccines, I am confident that public authorities decide in the best interest of the community.” 

*Complacency:* “Vaccination is unnecessary because vaccine-preventable diseases are not common anymore,” “My child’s immune system is so strong it also protects against diseases,” and “Vaccine-preventable diseases are not so severe that I should vaccinate my child.” 

*Constraints:* “Everyday stress prevents me from getting my child vaccinated,” “For me, it is inconvenient to have my child vaccinated,” and “Visiting the doctor makes me feel uncomfortable; this keeps me from having my child vaccinated.”

*Calculation:* “When I think about getting my child vaccinated, I weigh benefits and risks to make the best decision possible,” “For each and every vaccination, I closely consider whether it is useful for my child,” and “It is important for me to fully understand the topic of vaccination before I have my child vaccinated.”

*Collective Responsibility:* “When everyone is vaccinated, I don’t have to vaccinate my child, too,” “I have my child vaccinated so my child can also protect people with a weaker immune system,” and “Vaccination is a collective action to prevent the spread of diseases.”

*Added variables: Religion:* This was measured by “My religion does not support vaccination”. *Masculinity:* The measurement item was “My husband’s approval is important to vaccinate our child”. *Rumors or Misinformation:* this variable assessed conspiratorial and belief elements such as “Vaccination against HPV promotes premarital sex,” “Vaccination is a means to reduce our population,” and “The HPV vaccine is meant to make our girls unable to have children in future, and prayers are an effective way to prevent vaccine-preventable diseases.” 

*Vaccination intention:* The intention to vaccinate was measured by one item for RI: “I definitely intend to vaccinate my child when the next vaccination appointment is due” and one item for HPV “I will vaccinate my daughter against HPV in the future.” Responses ranged from “strongly disagree” to “strongly agree.” These two outcome variables tried to understand current behavior vis à vis future decision on childhood routine and HPV vaccination for children and adolescent daughters, respectively. 

## 3. Data Analysis 

The data were analyzed using R (version 3.6.3, TIBCO Software Inc., Seattle, USA). Multiple regression analyses were explored, upon which backward elimination regression analysis was performed to investigate if and how the 5C+ influenced the caregivers’ intention to vaccinate children for RI and HPV and their knowledge about HPV and cervical cancer.

### Ethical Considerations

Ethical clearance and approval from the National Health Sciences Research Committee (ref. no.: 20/04/2544) was obtained before commencing data collection. Authorization for the study was also received from the Malawian Ministry of Health (Ref. no. MED/1/3). Informed written consent was obtained from all subjects prior to their participation. 

## 4. Results

### 4.1. Demographic Characteristics

The participants were 18 years or older (18–24 years: 18%; 25–34 years: 40%; 35–45 years: 33%; 45–60 years: 8%; 60 years and older: 1%); 81% of participants were female, and 19% were male. The majority indicated being Christian (75%) or Muslim (22%). Few indicated believing in traditional religion (1%) or not being religious at all (2%). In terms of education, 11% had no formal education, 38% had completed primary education, 29% had completed secondary education and 4% had completed tertiary education.

### 4.2. Sources of Vaccination Information

About 82% of the respondents relied on healthcare workers for vaccination information, while 7% indicated friends and 4% family members as the most important source for vaccination information. Other sources, such as social media (Facebook, Twitter) and places of religious worship (mosque, church), were indicated by 1% or less. When asked about the accuracy of or trust in vaccination information provided by the different sources, trust in the community (*M* = 4.59, *SD* = 0.90) and healthcare workers (*M* = 4.30, *SD* = 1.32) was high. Trust in information received from friends and family (*M* = 1.64, *SD* = 1.23), social media (*M* = 1.32, *SD* = 0.89), and religious organizations (*M* = 1.65, *SD* = 1.25) was rather low.

### 4.3. 5C Psychological Antecedents of Vaccination

Each dimension of the original 5C included three items to be averaged. However, internal consistency was unexpectedly low for the five scales (see Appendix A, Figure A1), with Cronbach’s alpha ranging between −0.16 for collective responsibility and 0.4 for constraints. The maximum intercorrelation between two single items was r = 0.41 (for *visiting the doctor makes me feel uncomfortable* and *own religion does not support vaccination*). Consequently, we refrained from calculating any scale means and instead included all items separately in the subsequent analysis. Table 1 provides means (*M*) and standard deviations (*SD*s) for all items, including the religion, rumors, and masculinity extensions (5C+). 

### 4.4. Determinants of Caregiver’s Intention to Vaccinate Child

Explorative backward elimination regression analyses were performed to determine the relevant demographic and psychological variables related to vaccination intention. Backward elimination regression analysis is a stepwise regression approach that starts from a complete (saturated) model and at each step gradually eliminates variables from the regression model to find a reduced model that best explains the data. The approach is useful because it reduces multicollinearity problems and is a good means to resolve overfitting [70]. First, the 5C+ were considered predictors of the intention to vaccinate one’s child when the next appointment is due. The backward elimination regression algorithm removed those items that did not explain variance in intentions, resulting in a model with nine predictors (Table 2). Confidence in vaccine safety was the strongest predictor of vaccination intention, followed by the constraint of everyday stress. Stronger agreement that the topic of vaccination must be fully understood was associated with higher vaccination intentions, while items from the collective responsibility subscale did not play a role. For the complacency items, inconsistent results emerged. Vaccination intentions were lower for those thinking that the child’s immune system protects against diseases but higher for those believing that vaccine-preventable diseases are not severe. The importance of a husband’s approval increased vaccination intention.

Second, for all significant variables, backward elimination regression analyses were performed to explore factors predicting higher or lower levels of confidence, constraints, etc. Demographic variables such as age, gender, religion, education, and employment, as well as belief in rumors (about prayers preventing measles, vaccinations being a means to reduce the population, and HPV vaccine ruining fertility) and trust in healthcare workers were considered predictors (Table 3). Participants had lower confidence in vaccine safety when they believed in rumors and misinformation, such as prayers preventing measles and the HPV vaccine ruining fertility. Confidence in vaccine effectiveness was also affected by misinformation and was lower for unemployed participants but higher for those whose child was male and who had more trust in healthcare workers. For belief about the child’s immune system protecting against diseases, none of the given predictors was significant. On the contrary, thinking that vaccine-preventable diseases are not severe was affected by multiple variables: older adults (35–60 years), males, those believing in traditional religions, and those agreeing with the rumor that vaccination is a means to reduce the population. Participants whose child was male, who had at least secondary education, and who thought that prayers prevent measles indicated more strongly that everyday stress prevented them from vaccinating their child. Visiting the doctor made the participants feel more uncomfortable if they believed in traditional religion and misinformation (i.e., that vaccination is a means to reduce the population and that the HPV vaccine ruins fertility) and if they had lower trust in healthcare workers. Those who trusted healthcare workers also agreed with the statement that the topic of vaccination must be fully understood. A husband’s approval for vaccination was more important for participants who had primary or secondary education, more trust in healthcare workers, and stronger belief that vaccination is a means to reduce the population and that the HPV vaccine ruins fertility.

### 4.5. HPV Knowledge

Most participants (*n* = 521 vs. *n* = 67: 87% vs. 11%) indicated having heard about cervical cancer. About two-thirds had heard about human papillomavirus (*n* = 406 vs. *n* = 174: 68% vs. 29%) and knew that the cancer could be prevented (*n* = 420 vs. *n* = 159: 70% vs. 27%). However, only about one-fifth knew about the HPV virus (*n* = 130 vs. *n* = 453: 22% vs. 76%). By aggregating the four knowledge indicators, the mean HPV knowledge was 0.64 (*SD* = 0.30).

Again, a backward elimination regression analysis was performed to identify the variables related to HPV knowledge. Age, gender, religion, education, employment, belief in rumors or misinformation (i.e., believing that vaccination is a means to reduce the population, HPV vaccines reduce girls’ fertility, and vaccination against HPV promotes premarital sex), and trust in healthcare workers were considered predictors of mean HPV knowledge (Table 4). Religion and education were major drivers of HPV knowledge. Those believing in traditional African religion had lower HPV knowledge compared to those who were not religious, and respondents who had completed secondary or tertiary education showed higher knowledge than those without formal education. With regard to rumors or misinformation, believing that HPV vaccines impact fertility increased HPV knowledge.

### 4.6. Determinants of Intention to Vaccinate Daughters against HPV

A backward elimination regression analysis was performed to determine the demographic and psychological variables related to HPV vaccination intention. Age, religion, education, employment, gender, HPV knowledge, the 5C+, rumors or misinformation, and trust in healthcare workers were considered predictors of the intention to vaccinate daughters against HPV. Compared to the previous regression predicting intention to vaccinate the child, the gender of the child and the belief that prayers prevent measles were not included here, as the analyses focused on daughters above 9 years of age and HPV. For the analysis, respondents were excluded if they had no daughter above 9 years of age or if she had already received the HPV vaccine, resulting in a reduced sample of *n* = 133. 

Table 5 shows the standardized regression results. Of all the demographic variables, only education influenced vaccination intention. Respondents with a secondary or tertiary education showed lower intentions than those with no formal or primary education. HPV vaccination intentions increased with trust in healthcare workers but decreased with higher confidence in the safety of the vaccine and the system delivering it. When respondents thought that their children’s immune systems protected against diseases, vaccination intentions increased. Similar effects could be observed for indicators of calculation and collective responsibility: weighing risks and benefits and thinking that vaccination protects people with weaker immune systems increased intentions to have adolescent daughters vaccinated against HPV. Husbands’ approval also had a positive effect on vaccination intentions. Interestingly, for knowledge about HPV and its prevention, no direct effect on vaccination intentions could be found.

### 4.7. Belief in Rumors

Since belief in rumors and misinformation (i.e., that prayers prevent measles, vaccines are a means to reduce the population, and the HPV vaccine ruins girls’ fertility) was associated with both general intentions to vaccinate children and HPV knowledge, an additional backward elimination regression was performed to identify the characteristics associated with those beliefs. A mean belief in rumors was first calculated from the three items (*M* = 2.46, *SD* = 1.17, Cronbach’s alpha = 0.67). Age, gender, religion, education, employment, and trust in healthcare workers, social media, religion, friends, family, and the local community were considered predictors of mean belief. As indicated by the standardized regression results (Table 6), age was a significant predictor: compared to the youngest group of participants (18–24 years), respondents aged 25–34 years old indicated a higher belief in rumors. For older participants, beliefs decreased, with the oldest participants (60+ years) showing the least belief. Being unemployed increased the average belief in rumors, while trust in healthcare workers reduced it. No effects could be found for religion, education, gender, and trust in other sources of information, such as social media or friends and family.

## 5. Discussion

Based on the study outcomes, caregivers’ vaccination acceptance for RI is motivated predominantly by aspects related to confidence in vaccine safety, followed by everyday stress (constraints; an unexpected relation) and some minor influences by variables, such as vaccine effectiveness (confidence), beliefs in the child’s immune system (complacency), or husband’s approval (masculinity). Other variables that correlated with the intention to vaccinate one’s child were the feeling that vaccine-preventable diseases are not severe (complacency), fear of doctors (constraints), and risk-benefit analysis and understanding vaccine topics (calculation). 

In sum, confidence in vaccine safety was the strongest predictor for vaccination intention. Confidence in vaccines did not differ according to demographic characteristics but was strongly related to rumors (i.e., that prayers prevent measles; and HPV vaccine ruins fertility). A further analysis demonstrated that confidence in vaccine safety decreased when people believed that prayers or religious rituals (e.g., taking “holy communion”) could prevent or serve as prophylaxis against vaccine-preventable diseases, such as measles. 

Confidence in vaccine safety and effectiveness are global vaccine hesitancy phenomena, especially in high-income countries where studies and countermeasures are exhaustive [44,71]. However, low-income settings, such as Malawi, will presumably require a more bottom-top approach, since the majority of its population lives in rural areas. Additionally, since the caregivers in this study trusted healthcare workers most for their vaccination information, the expanded program on immunization (EPI) in Malawi should integrate local healthcare workers into the heart of vaccination education.

It is not just the safety of vaccines that worries caregivers about RI but also whether vaccines are effective. Assessing this alongside other variables showed that confidence in vaccine effectiveness increased with trust in vaccination information provided by healthcare workers but decreased if the participants were unemployed or thought that the HPV vaccine reduces fertility. For the HPV vaccine, decreased intention to vaccinate a daughter against cervical cancer was traced to belief in rumors and the caregiver’s lack of confidence in the ability of the public authorities to decide in the best interest of the populace. 

Belief in rumors and being unemployed had a negative effect on safety perception; therefore, believing in these rumors and not having a job both decreased vaccination intention, hence increasing vaccine hesitancy. Addressing vaccine hesitancy based on rumors by using health promotion campaigns to increase vaccination demand requires a well-tailored and specific communication strategy. The strategy should not only debunk rumors surrounding vaccines, such as HPV but also resolve vaccine safety and effectiveness concerns raised by caregivers. Additionally, since unemployment plays a significant role in vaccine hesitancy, incentives could be built around vaccination attendance at health facilities, in form of provision of lunch and transportation reimbursement for caregivers, besides maintaining a free vaccination program. Incentivizing unemployed caregivers will mitigate constraints that accompany out-of-pocket costs during vaccination. This could help to avoid complacency and dropout. 

Complacent behavior among caregivers was a unique finding. The perception that vaccine-preventable diseases are not so severe or not perceiving diseases as high risk and vaccination as necessary was predominantly found among the older population (35–60) compared to the young adults (25–34). One would expect that the older people become, the higher their health-seeking behavior or self-efficacy attached to preventive behavior and vulnerability to diseases [72,73]. This is also contrary to expectations that younger individuals have higher risk-seeking and invulnerability behavior due to feelings of positive subjective personal health status [72,74,75]. As complacency correlates with a positive general risk attitude, risk-seeking behaviors are usually expected [49,76]. However, it is remarkable to find this among the older population instead of the younger. Perhaps they thought that these childhood diseases are not relevant for them. Further studies may be needed to understand the reasons for this trend, and a targeted intervention at the older population on the severity of vaccine-preventable diseases, preferably using negative framing, is recommended. 

The role of education was notable because education mostly correlates with positive attitudes and behavior toward vaccination [77,78,79]. The opposite, as in the case of this study, is not frequently found. However, Malawi may not necessarily be the only country with this finding. The influence of education on childhood immunization uptake in Spain was found to be unrelated: the less educated parents had higher childhood immunization rates [80]. Similarly, when a state of emergency was declared in Washington state in 2019 due to a measles outbreak, affluent and well-educated parents were most hesitant to vaccinate children [81]. Therefore, communication messages that target risk perception among urban population is an appropriate intervention. 

Rumors and misinformation has hitherto been under-estimated in various vaccine demand creation interventions across the SSA region [69,82], and it is an important revelation of this study. There was a strong relationship between rumors and vaccination intention among caregivers for RI and HPV knowledge, and belief in rumors was a major barrier against HPV vaccination among caregivers of adolescent girls in Malawi. A closer analysis showed that being young adults, unemployed, or having low trust in healthcare workers increased those beliefs. The devastating impacts of rumors on vaccination demand cannot be overstated. Although being unemployed and having low trust in healthcare workers as drivers for low vaccination intention is not unexpected, in view of the social baggage that comes with it when analyzed. However, being young adults contributing to higher belief in rumors is a surprise and needs to be further investigated. Although several factors could influence this, including young people’s exposure to social media misinformation campaigns, where the bulk of vaccination misinformation is currently peddled [83,84,85,86]. 

A tailored strategy that employs scientific research and communicating them in an evidence-informed manner to inspire recurrent stakeholder dialogue by raising different voices to allow for discussions is needed. This process builds partnerships using facts and, at the same time, counters misinformation with a non-aggressive posture. Therefore, while social media use has been associated with a negative impact on public perception of vaccines and vaccinations, it also presents a unique opportunity to aggressively disseminate scientific evidence through the same platform. This could be a good way that young vaccine misinformation spreaders can be reached and convince using evidence. Additionally, although knowledge deficits arising from lack of cognitive information or affective or psychomotor abilities needed for positive health-seeking behavior seem apparent in Malawi, based on the results, when knowledge is confounded by education (especially little or no formal education), improving knowledge deficits among this group is not enough. Intervention geared toward addressing this must feel appropriate and fit people’s value systems and cultural norms. 

Trust is the bedrock of vaccination acceptance. Therefore, trust must be built in and around all facets of national EPI programs, including vaccine development, distribution, policies, healthcare systems (doctors, nurses, and immunizers), and mass vaccination campaigns. For vaccination acceptance strategies to succeed, efforts should focus first on building trust, improving vaccine confidence, and dispelling rumors associated with vaccines or vaccinations. 

Husbands or fathers were identified in this study as crucial to childhood vaccination uptake. A father’s consent for childhood vaccination had a significant positive effect on vaccination intentions. Even more so, a husband’s or father’s approval is important for women with little education (primary or secondary education), little trust in healthcare workers, and those who believe in rumors. Thus, interventions should focus on improving men’s positive perceptions of vaccines in Malawi. Encouraging men to attend immunization activities, like ante-natal initiatives, would be a positive step toward improving perception. 

Similarly, the role of gender on vaccination coverage have raised scholarly debate lately in LIMC. Disparities in immunization coverage between boys and girls have shown in some instances how gender of child determines uptake [87,88,89]. This study provides valuable insights on this debate. In Malawi, confidence in the effectiveness of vaccines is lower among caregivers whose children are female, compared to the male; hence, affecting vaccination behavior. Therefore, a female child in Malawi is more likely to delay, refuse or have an incomplete vaccination coverage. A lot more needs to be done to address gender gap and masculinity syndrome, especially in rural Malawi. 

Potential bias in the study might be associated with convenient sample of participants, therefore, future research should explore a more representative paradigm. Additionally, the low internal consistency observed in the 5C scale may be largely due to wordings, grammar, and language used for the constructs. Future studies should rephrase or translate the scale into simplified English language or local language, for optimal comprehension. Lastly, while the study tries to adapt, validate and use the 5C model to understand and measure vaccine hesitancy in non-western setting such as Malawi, it however, acknowledged the limitations associated with the cross-sectional method used. 

## 6. Conclusions

Overall, the 5C psychological antecedents model for measurement of vaccine hesitancy in Malawi was effective, at least for assessing RI compared to HPV vaccine, although not completely adaptable, as it was initially developed and used in high-income-countries. The results, plus its extension show that items measuring aspects of confidence, complacency, constraints, calculation, and masculinity were the significant factors that drive low vaccination uptake for RI. For the HPV vaccination, rumors, lack of trust in government (confidence), education level, and husband’s approval to vaccinate daughters played predominant roles. Thus, addressing vaccine hesitancy in Malawi requires a multi-dimensional approach that involves both communication tools and devolution of management of vaccination programs to the local level, led by local healthcare workers, since trust in them is high among the population. Further studies may be needed to understand low-risk perceptions among the older population and the potential role of local non-governmental organizations in building trust in vaccines. Additionally, further studies are necessary to understand low risk perception among elderly people, gender discrepancy, and the declining trend in second vaccine doses.

## Figures and Tables

**Table 1 vaccines-09-01231-t001:** Mean agreement with 5C+ antecedents of vaccination, per item.

Variable	*M*	*SD*
*Confidence*		
Vaccines are safe (Conf 1)	4.19	1.27
Vaccinations are effective (Conf 2)	4.30	1.25
Public authorities decide in the best interest of the community (Conf 2)	4.06	1.32
*Complacency*		
Vaccination-preventable diseases are not common anymore (Comp 1)	1.94	1.26
Immune system protects against diseases (Comp 2)	2.66	2.64
Vaccine-preventable diseases are not severe (Comp 3)	1.91	1.20
*Constraints*		
Everyday stress prevents vaccination (Cons 1)	2.47	1.49
Receiving vaccinations is inconvenient (Cons 2)	2.40	1.40
Visiting the doctor makes me feel uncomfortable (Cons 3)	2.05	1.20
*Calculation*		
Weighing benefits and risks (Calc 1)	3.52	2.70
Considering usefulness of vaccination (Calc 2)	3.97	1.39
Topic of vaccination must be fully understood (Calc 3)	4.32	1.20
*Collective responsibility*		
No need for vaccination when everyone is vaccinated (Coll 1)	1.96	1.21
Vaccinated to protect people with weaker immune system (Coll 2)	2.45	1.50
Vaccination as collective action to prevent the spread of diseases (Coll 3)	4.33	1.57
*Religion*		
Own religion does not support vaccination	1.88	1.02
*Masculinity*		
Husband’s approval important for vaccination	3.19	1.56
*Rumors*		
Vaccination causes infertility	2.40	1.48
Prayers prevent measles	2.76	1.54
Vaccination is means to reduce population	2.22	1.50

*Note*: Agreement was assessed using 5-point scales ranging from 1 = “strongly disagree” to 5 = “strongly agree.” *M*, mean. *SD*, standard deviation.

**Table 2 vaccines-09-01231-t002:** Backward elimination regression results for caregiver’s intentions to vaccinate children.

Predictor	β	b	SE	CI−	CI+
(Constant)		−0.43	0.38	−1.180	0.322
Confidence: Vaccines are safe	**0.52**	**0.64**	0.05	0.544	0.737
Confidence: Vaccinations are effective	**0.10**	**0.14**	0.05	0.035	0.236
Complacency: Child’s immune system protects against diseases	**−0.12**	**−0.06**	0.02	−0.105	−0.022
Complacency: Vaccine-preventable diseases are not severe	**0.10**	**0.13**	0.06	0.029	0.249
Constraints: Everyday stress prevents vaccination	**0.16**	**0.17**	0.04	0.087	0.260
Constraints: Visiting the doctor makes me feel uncomfortable	**−0.08**	**−0.11**	0.05	−0.210	−0.002
Calculation: Weighing benefits and risks	−0.05	−0.03	0.02	−0.071	0.010
Calculation: Topic of vaccination must be fully understood	**0.09**	**0.14**	0.06	0.025	0.247
Masculinity: Husband’s approval important for vaccination	**0.08**	**0.09**	0.04	0.009	0.167

*Note.* R^2^ = 0.41; adjusted R^2^ = 0.40. Bold values are significant at *p* < 0.05. b refers to the unstandardized estimate and β is the standardized version. SE refers to the standard error. CI− and CI+ are the lower and upper bounds of the 95% confidence intervals.

**Table 3 vaccines-09-01231-t003:** Backward elimination regression results predicting relevant psychological antecedents of vaccine hesitancy.

Predictor	Confidence		Complacency		Constraints		Calculation	Masculinity
	Vaccines are safe	Vaccines are effective	Child’s immune system protects against diseases	Vaccine-preventable diseases are not severe	Everyday stress prevents vaccination	Visiting the doctor makes me feel uncomfortable	Topic of vaccination must be fully understood	Husband’s approval important for vaccination
(Constant)	**4.97 (0.14) *****	**4.01 (0.27) *****	**2.68 (0.14) *****	**1.15 (0.42) ****	**2.62 (0.61) *****	**1.86 (0.47) *****	**3.78 (0.18) *****	**1.33 (0.65) ***
Age (Baseline: 18–24)								
25–34				–0.26 (0.16)				
35–44				**−0.51 (0.17) ****				
45–60				**−0.75 (0.25) ****				
60+				−0.77 (0.66)				
Gender: male (Baseline: female)				**0.40 (0.15) ****				
Gender of child: male (Baseline: female)		**0.36 (0.12) ****			**0.28 (0.14) ***			
Religion: (Baseline: not religious)								
Christian				0.66 (0.40)	−0.84 (0.51)	0.13 (0.41)		−0.09 (0.54)
Muslim				0.75 (0.41)	−0.31 (0.53)	0.26 (0.43)		0.30 (0.56)
Traditional				**1.79 (0.63) ****	−0.09 (0.82)	**1.71 (0.67) ***		1.34 (0.88)
Other				0.19 (1.17)	−2.41 (1.52)	−1.09 (1.24)		−2.51 (1.62)
Education (Baseline: no formal education)								
Primary education					0.42 (0.22)			**0.48 (0.23) ***
Secondary education					**0.62 (0.23) ****			**0.63 (0.25) ***
Tertiary education					**1.14 (0.46) ***			0.90 (0.49)
Being unemployed		**−0.27 (0.12) ***		−0.22 (0.11)		−0.22 (0.12)		0.25 (0.16)
Trust in healthcare workers		**0.09 (0.05) ***			−0.10 (0.05)	**−0.10 (0.05) ***	**0.14 (0.04) *****	**0.14 (0.06) ***
Believing in rumors:								
Vaccination is means to reduce population				**0.18 (0.04) *****		**0.12 (0.04) ****		**0.12 (0.05) ***
Prayers prevent measles	**−0.11 (0.04) ***	0.07 (0.04)			**0.13 (0.05) ****			
HPV vaccine ruins fertility	**−0.22 (0.05) *****	**−0.16 (0.05) *****				**0.12 (0.04) ****		**0.14 (0.06) ***

*Note.* Estimates: b, SE in parentheses. Bold estimates: *p* < 0.05. Significance indicators: * 0.05 > ** ≥ 0.01 > *** ≥ 0.001.

**Table 4 vaccines-09-01231-t004:** Standardized regression results predicting mean HPV knowledge.

Predictor	β	b	SE	CI−	CI+
(Constant)		**0.64**	0.10	0.44	0.84
Gender: male (Baseline: female)	−0.07	−0.05	0.03	−0.11	0.02
Religion (Baseline: not religious)					
Christian	−0.15	−0.10	0.10	−0.29	0.09
Muslim	−0.25	−0.18	0.10	−0.38	0.01
Traditional	**−0.18**	**−0.43**	0.14	−0.71	−0.16
Other	0.00	0.00	0.22	−0.44	0.44
Education (Baseline: no formal education)					
Primary education	0.08	0.05	0.04	−0.03	0.13
Secondary education	**0.28**	**0.18**	0.04	0.10	0.27
Tertiary education	**0.11**	**0.17**	0.08	0.02	0.32
Believing in rumor: HPV vaccine ruins fertility	**0.10**	**0.02**	0.01	0.00	0.04

*Note.* R^2^ = 0.09; adjusted R^2^ = 0.07. Bold values are significant, with *p* < 0.05. b refers to the unstandardized estimate and β is the standardized version. SE refers to the standard error. CI− and CI+ are the lower and upper bounds of the 95% confidence interval.

**Table 5 vaccines-09-01231-t005:** Standardized backward elimination regression results predicting intentions to have adolescent daughters vaccinated against HPV.

Predictor	β	b	SE	CI−	CI+
(Constant)		**2.93**	1.07	0.83	5.03
Age (versus 18–24)					
25–34	−0.30	−1.08	0.73	−2.50	0.35
35–44	−0.09	−0.32	0.73	−1.76	1.12
45–60	−0.07	−0.38	0.80	−1.96	1.20
Education (versus no formal education)					
Primary education	−0.21	−0.74	0.44	−1.61	0.12
Secondary education	**−0.33**	**−1.25**	0.48	−2.19	−0.31
Tertiary education	**−0.22**	**−4.56**	1.61	−7.72	−1.40
Trust in healthcare workers for accurate vaccination information	**0.19**	**0.24**	0.10	0.05	0.43
Confidence: Vaccines are safe	**−0.17**	**−0.21**	0.10	−0.41	0.00
Confidence: Public authorities decide in the best interest of the community	**−0.24**	**−0.32**	0.11	−0.53	−0.11
Complacency: Immune system protects against diseases	**0.24**	**0.30**	0.10	0.11	0.50
Calculation: Weighing benefits and risks	**0.24**	**0.28**	0.09	0.11	0.45
Collective responsibility: Vaccinate to protect people with weaker immune system	**0.27**	**0.31**	0.09	0.13	0.49
Masculinity: Husband’s approval important for vaccination	**0.27**	**0.23**	0.09	0.06	0.40

*Note.* R^2^ = 0.34; adjusted R^2^ = 0.26. Bold values are significant with *p* < 0.05. b refers to the unstandardized estimate and β is the standardized version. SE refers to the standard error. CI− and CI+ are the lower and upper bounds of the 95% confidence interval.

**Table 6 vaccines-09-01231-t006:** Standardized backwards regression predicting average belief in rumors.

Predictor	β	b	SE	CI−	CI+
Age (versus 18–24)		**3.18**	0.20	2.79	3.57
25–34	**0.15**	**0.36**	0.13	0.09	0.62
35–44	0.10	0.24	0.14	−0.04	0.51
45–60	−0.05	−0.22	0.20	−0.61	0.18
60+	**−0.09**	**−1.36**	0.64	−2.62	−0.09
Being unemployed	**0.10**	**0.25**	0.10	0.06	0.44
Trust in healthcare workers for accurate vaccination information	**−0.28**	**−0.24**	0.03	−0.31	−0.17

*Note.* R^2^ = 0.13; adjusted R^2^ = 0.12. Bold values are significant with *p* < 0.05. b refers to the unstandardized estimate and β is the standardized version. SE refers to the standard error. CI− and CI+ are the lower and upper bounds of the 95% confidence interval.

## Data Availability

The materials and dataset used or analyzed are available at the Open Science Framework https://osf.io/pzaer/ (accessed on 20 October 2021).

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
