# Peer review of "Caregivers’ Willingness to Vaccinate Their Children against Childhood Diseases and Human Papillomavirus: A Cross-Sectional Study on Vaccine Hesitancy in Malawi"

_vaccines, 2021, doi:10.3390/vaccines9111231_

Round 1

Reviewer 1 Report

The article titled "Caregivers’ Willingness to Vaccinate their Children Against Childhood Diseases and Human Papillomavirus: A Cross-Sectional Study on Vaccine Hesitancy in Malawi" by Adeyanju, G.C. et. al. focusses on understanding some of the factors that can potentially contribute to hesitancy in HPV vaccine adoption and adherence in Malawi.

The authors used multiple regression analyses to explore vaccination intention predictors on sample populations derived from three separate districts representing variable vaccine uptake- Dowa with low percentage and Salima with high uptake numbers. Zomba, where HPV vaccination has already been piloted was also included. The cohort consisted of 600 participants. 5C+ model to compute vaccine hesitancy. In addition to the conventionally used 5 variables- confidence, complacency, constraints, calculation, and collective responsibility; this model utilizes religion, rumors and masculinity to further strengthen the assessment. 

The study concluded that confidence in vaccine safety was consistently among the strongest predictors of vaccine hesitancy. The results based  on 5C+ are concisely tabulated. The authors discussed each variable and provide an excellent Malawian context to the results obtained. Using the results obtained, the authors conclude that the 5C model is quite effective to measure vaccine hesitancy in countries like Malawi, although it's not completely adaptable. Individual variables are discussed along with some unexpected findings and suggestions to improve vaccination strategy. 

The study hypothesizes and suggests a tailored strategy utilizing a multidimensional approach and also highlights importance of non-governmental organizations in helping build trust in vaccines. 

Overall, I believe the article is highly relevant and the study is very well presented.

Author Response

POINT-BY-POINT-RESPONSE-TO REVIEWER’S-COMMENTS

REVIEWER 1

S/NO

REVIEWER’S COMMENTS

AUTHOR’S RESPONSES

1

Spelling check

We have carried out extensive check

2

Methods section can be improved.

We re-wrote large parts of the Methods section.

Reviewer 2 Report

Review of “Caregivers’ willingness to vaccinate their children again childhood diseases and Human Papillomavirus: a cross-sectional study of vaccine hesitancy in Malawi.”

This is a valuable article on vaccine hesitancy in Malawi and I endorse publication of it, conditional on revisions to make presentation of results more transparent and easy to read.  Vaccine hesitancy in Africa is an important topic in general, not well understood, and growing in significance with the Covid-19 pandemic.  This work could not be more timely.  Its effort to understand the role of rumors is particularly interesting and innovative.  While the work is descriptive and correlational rather than causal (perhaps something the authors should say up front), our general low level of information on the topic warrants descriptive work.

Aside from missing an Introduction, the article was well written.  The results tables were difficult to follow, however, and there were a few issues with the discussion of variables, especially the outcome variables, that need to be addressed. 

  • The authors should consider including a standard Introduction to the paper. The “Background” section comes first, which is confusing, because it discusses general trends and other vaccine campaigns.  Why are trends in DPT3 the first thing we are learning in a paper about HPV vaccination in Malawi?  In the Introduction, the authors might acknowledge the limitations of their cross-sectional method – descriptive, correlational, not causal. 
  • What kind of bias might be introduced by the convenience samples?
  • Are boys and girls vaccinated against HPV in Malawi?
  • Some visual representations of the descriptive data would be nice. For example see: https://www.mdpi.com/2076-393X/9/7/772/htm
  • Table 1: it is a bit odd not to align the text in the tables to the left (rather than centering it).
  • “internal consistency was unexpectedly low for the five scales.” What should we make of this?  Are the scales not relevant for Malawi?  I would like to learn more about the correlations across scale items as we know so little about vaccine hesitancy in Malawi, this descriptive information is useful.  If space is an issue, it could go in an appendix.
  • Clarify: what is “b” in Table 2 (also correct alignment so it is level with other parameters). Same comment for Table 4. (In my field, we do not use “backward elimination regression analysis,” so even if this presentation is standard for this method, it is not familiar to me and possibly other readers). 
  • Why are demographic factors like education not included in Table 2, intention to vaccinate children?
  • Table 3 is difficult to read. Perhaps it should be broken into separate tables for each outcome?  There are too many outcomes arrayed across the columns to work as a single table.  Perhaps coefficient plots could be used to visually represent the data instead of tables?  Also, why use stars here to indicate significance when earlier bolding was used? 
  • The discussion of Table 3 is interesting but could use some streamlining or organizing to make it easier to follow. It is hard to make sense of the patterns when there are so many outcomes and variables being discussed.
  • Why would the gender of the youngest child matter?
  • There is a lack of clarity on the first outcome variable. Through most of the text, it is described as “intention to vaccinate child at next appointment.” But then in the section where the analysis shifts to the second outcome variable (“intention to vaccinate daughter”), the first outcome variable then shifts to “intention to vaccinate youngest child.”  (See line 374). In general, greater attention to the outcome variables, what they actually are, and why they were used, would be useful. 
  • It is puzzling that there are so many differences across outcome variables, and in general, the first outcome variable (intention to vaccinate child at next appointment) seems to behave more as expected (i.e. to be predicted by the set of variables the authors think are important). Why is that? Again, greater attention to the difference between these two outcomes and why they are predicted by different factors would be helpful.
  • The section “belief in rumors” is quite interesting and innovative. Definitely a strength of the article. Why would there be a non-linear relationship for age?  The authors summarize in the conclusion that the youth are more likely to believe in misinformation and rumors but in fact the youngest group are less likely and it is the 25-34 group that is most likely.  Do those under 25 not have phones perhaps?
  • In the conclusion, the authors mention the counter-intuitive findings on education. This is interesting and important. I missed it on the first reading (line 380).  Perhaps more attention should be paid there.  Also, in the conclusion, the authors should clarify that this is only the “vaccinate daughters” outcome.  They do not estimate education for the first outcome variable (why?). 
  • Would it be possible to illustrate with simple graphics some of the chains of correlations the authors are uncovering? (space permitting). Such as:

Being a young adult (age 25-34)  => belief in rumors => belief vaccines not safe/not effective => vaccine hesitancy

Reviewer 3 Report

The manuscript by Adeyanju et al. addresses vaccine hesitancy among parents/guardians of children and young girls in Malawi towards routine immunization and HPV vaccination.
I believe that the topic is of great interest to the scientific community, just as I believe it is essential to produce further scientific knowledge on the subject of VH in SSA countries.
The insights are various and very sound. Among all, the study of the variable on "masculinity", rarely investigated in studies conducted for example in the USA or EU.

However, I believe that the manuscript needs an extensive revision before being considered suitable for publication.

Abstracts:
L21-L22 please indicate vaccine uptake in %, avoiding the use of "low" or "high".
The abstract needs to be rewritten in its entirety to make it smoother, especially L30-35.

Background:
Similarly, also in L83-85 please provide data instead of "often", "most". Same as L101.
L107-109. By when is the vaccination of 1.5 million adolescents planned?
L167-192. I think the explanation of the 5C (adjusted) model needs to be summarised and placed within the methods section explaining how the study variables were selected.

Methods:
L195-206: I would not list the original intentions of the study but only what was actually carried out. Please indicate, as requested above, the vaccine uptake percentages of the districts, as they were chosen based on this.
L226-231: I would include here the section of the 5C+ ( summarised), in order to motivate the choice of variables used.
L250, L254, L255, L263: Please specify how many items make up the scale.

Results:
L276-277: Please also add the percentage of F and M participants in the study.
L288-290: In this specific point and, in general, in the whole section of the results, I recommend avoiding the authors' statements and just presenting the results (e.g. avoiding the use of "high", "rather low", etc.).
Table 2: in the notes also the explanation of b, B, SE. Same in the other tables.
L304-309; L320-324; L369-378; L396-403: the methodology of how the analyses were conducted should be explained in details in the methods section.

Discussion:
I recommend making the discussion smoother and more focused on the main findings of the study.
For example, I suggest summarising the first part of the discussion by removing repetitions from the introduction section.

In general, the discussion as well as the conclusions are supported by the findings and useful for the scientific discussion on VH.

Round 2

Reviewer 3 Report

I thank the authors for their answers in the attached file. However, the tracked version does not include all the requested changes. I do not know whether this is a technical problem with loading the revised version or whether the changes have not been actually made.

For example, the authors report that they have modified the abastract by addressing terms such as "high" and "low" in a more specific way. These changes are not only not highlighted, they are not even present. I don't think "many" is specific enough for a data presented in a scientific paper.

In answer no.3 the authors explain that they have used more narrative terms in order to be more informative. Well, the journal they have chosen is a scientific journal which is aimed at professionals and researchers.

Author Response

POINT-BY-POINT-RESPONSE-TO REVIEWER’S-COMMENTS (SECOND ROUND) 

REVIEWER 3

S/NO

REVIEWER’S COMMENTS

AUTHOR’S RESPONSES

the tracked version does not include all the requested changes. I do not know whether this is a technical problem with loading the revised version or whether the changes have not been actually made

For example, the authors report that they have modified the abastract by addressing terms such as "high" and "low" in a more specific way. These changes are not only not highlighted, they are not even present. I don't think "many" is specific enough for a data presented in a scientific paper.

We apologize for that. While most of us were on research retreat, the only author who was not, had an initial check specifically on the quantifiers (“many” “high”, “low”) and forgot to leave the track changes for TWO corrections. Notably, in Line 103 “most” was replaced with 70%, in Line 115 “low” was replaced with “below 90%”.

“The assessment showed that many children in the two districts had an incomplete number of doses) (Line 86-87)”. This is a reference to the preceding sentence, which already provided the data. This is similar to how most of the quantifiers are used in the manuscript.

In answer no.3 the authors explain that they have used more narrative terms in order to be more informative. Well, the journal they have chosen is a scientific journal which is aimed at professionals and researchers.

Text descriptions or narratives in this manuscript were derived from the scientific calculations provided in the tables. “Many” “higher” “low” “most” were mere description or emphasis of the already presented scientific data.

Many thanks for the feedback.

Round 3

Reviewer 3 Report

I thank the Authors for clarifying the points of the previous revision. I believe that the manuscript still has some flaws, but not decisive enough to rule out publication.